# Impact of Nutritional Changes on the Prognosis in Pancreatic Cancer Patients Underwent Curative Surgery After Neoadjuvant Chemotherapy

**DOI:** 10.3390/nu17040647

**Published:** 2025-02-11

**Authors:** Seulah Park, Go-Won Choi, Inhyuck Lee, Younsoo Seo, Yoon Soo Chae, Won-Gun Yun, Youngmin Han, Hye-Sol Jung, Wooil Kwon, Joon Seong Park, Jin-Young Jang, Young Jae Cho

**Affiliations:** Department of Surgery and Cancer Research Institute, Seoul National University College of Medicine, 101 Daehak-ro, Jongno-gu, Seoul 03080, Republic of Korea; naandzzaya@gmail.com (S.P.); reve7373@gmail.com (G.-W.C.); gsrcu@naver.com (I.L.); ys_seo93@naver.com (Y.S.); codbstnsla@gmail.com (Y.S.C.); wkeonyun5@naver.com (W.-G.Y.); vickijoa@gmail.com (Y.H.); hyesoljung@gmail.com (H.-S.J.); willdoc78@gmail.com (W.K.); jspark330@gmail.com (J.S.P.); jangjy4@snu.ac.kr (J.-Y.J.)

**Keywords:** pancreatic cancer, neoadjuvant chemotherapy, nutritional status, survival rate

## Abstract

Background: Pancreatic cancer is a highly aggressive malignancy with a poor prognosis. Neoadjuvant chemotherapy (NAC) is increasingly used to improve survival in patients with pancreatic cancer; however, it often results in nutritional deterioration, which may negatively impact patient outcomes. Therefore, this study aimed to assess the effect of changes in nutritional status on the long-term outcomes of patients with pancreatic cancer who underwent curative surgery after NAC. Methods: This retrospective single-center study included 148 patients with pancreatic cancer who underwent curative surgery after NAC between 2010 and 2020. The Controlled Nutritional Status (CONUT) score was used to determine the nutritional status of the patients. Patients were categorized into worsened, maintained, and improved groups based on the changes in their CONUT scores before and after NAC. We compared differences in overall survival (OS) and disease-free survival (DFS) between the groups. Results: The worsened nutritional status group exhibited the shortest median OS (28 months) compared to the maintained and improved groups (39 and 66 months, respectively; *p* = 0.01). Additionally, the worsened group demonstrated the shortest DFS compared to the other two groups (13, 22, and 39 months, respectively; *p* = 0.02). Multivariate analysis identified nutritional deterioration as an independent prognostic factor for OS (hazard ratios (HR), 2.11; 95% confidence intervals (CI), 1.31–3.40; *p* < 0.01). Conclusions: Nutritional deterioration after NAC is a significant prognostic factor of poor survival outcomes in patients with pancreatic cancer. These findings indicate that serial nutritional assessments and treatment during NAC are crucial for improving patient outcomes.

## 1. Introduction

Pancreatic cancer is currently the third leading cause of cancer-related deaths in the United States and is anticipated to become the fourth leading cause of cancer-related mortality by 2040 [1,2]. However, at the time of diagnosis, only 10–20% of patients with pancreatic cancer are considered for surgical resection because they are unresectable or in a metastatic state [3,4,5].

Recent studies indicate that neoadjuvant chemotherapy (NAC) has shown promise in improving survival outcomes for patients with locally advanced pancreatic cancer (LAPC) and borderline resectable pancreatic cancer (BRPC), with better survival outcomes relative to those who underwent surgery first [3,6,7,8]. Given these benefits, upfront chemotherapy has been increasingly adopted in recent years for the treatment of pancreatic cancer.

However, NAC has been reported to have many side effects, particularly the worsening of the nutritional status of patients [9]. Numerous studies have indicated that malnutrition may enhance chemotherapy-related toxicities, reduce treatment tolerance, and even increase surgical complications [10,11]. Furthermore, pancreatic cancer itself has been linked to pancreatic exocrine insufficiency; it results in unintentional body weight loss (BWL) that may occur as a result of anorexia, malabsorption and/or cachexia [12,13]. Up to 85% of patients with pancreatic cancer present with BWL at diagnosis, and approximately 50% experience preoperative BWL, 80% will develop progressive BWL following diagnosis [14,15,16]. Currently, no approved medical therapies exist for managing pancreatic cancer-associated BWL. This deterioration in nutritional status including BWL not only affects the quality of life and physical activity of patients with pancreatic cancer but also affects their overall survival (OS) [17,18,19,20,21,22].

As the use of NAC continues to increase in the treatment of pancreatic cancer, the nutritional impact of cancer treatment is well known. However, most existing studies have focused on a single time point, such as preoperative status or early treatment phases, and the effects of nutritional treatment on outcomes. Nevertheless, these studies have not comprehensively addressed the longitudinal changes in nutritional status before and after NAC [23,24]. Few studies have specifically investigated how these longitudinal changes in nutritional status over the course of NAC influence patient prognosis. Given that poor nutritional status is known to compromise treatment tolerance, quality of life, and survival, a clearer understanding of the timing and extent of nutritional changes is essential. Therefore, this study aims to investigate the effect of nutritional changes before and after NAC on the prognosis of pancreatic cancer patients.

## 2. Methods

### 2.1. Study Design and Patients

Growing evidence supports the critical role of nutritional status in patients with cancer because it can predict tumor development and treatment tolerance [25]. Traditionally, inflammation-based markers, such as the prognostic nutritional index (PNI), Glasgow prognostic score, C-reactive protein-to-albumin ratio, and geriatric nutritional risk index, have been identified as effective prognostic factors for various cancers [26,27,28,29,30,31]. Historically, the PNI, a validated scoring tool reflecting immunonutritional status, has been widely used to estimate treatment tolerance and cancer progression [32]. More recently, the Controlling Nutritional Status (CONUT) score, another indicator of immunonutritional status, was introduced as a tool for screening the nutritional status of patients by Ignacio et al. [33]. As an immune–nutritional biomarker, the CONUT score is measured by serum albumin (ALB), total lymphocyte count (TLC), and total cholesterol (TC) concentration (Table 1).

It has been recognized as a prognostic factor for pancreatic cancer and other malignancies in patients undergoing chemotherapy or surgical resection [34,35,36,37,38]. Additionally, the CONUT score is based on its demonstrated superiority in predicting prognosis compared to other nutritional and inflammatory markers. Previous studies have shown that the CONUT score offers a more comprehensive and reliable assessment of immune–nutritional status. Notably, CONUT score has exhibited a higher sensitivity and predictive accuracy in predicting patient outcomes, as evidenced by higher AUC values in comparison to other scoring systems [39]. To further explore the impact of nutritional status, survival analysis was also conducted based on BWT changes and the PNI, which reflect both nutritional and immune status. To further explore the impact of nutritional status, survival analysis was also conducted based on BWT changes and the PNI, which reflect both nutritional and immune status.

In contrast to previous studies that focused on single time points, such as preoperative status or early treatment phases, this study incorporates a longitudinal approach, tracking changes in nutritional status before and after NAC. This comprehensive approach allows for a more thorough understanding of the nutritional dynamics over the course of treatment in pancreatic cancer research. By examining these longitudinal changes in nutritional status using the CONUT score, this study aims to clarify how fluctuations in nutritional status influence prognosis and treatment outcomes in pancreatic cancer patients.

The Institutional Review Board of Seoul National University Hospital approved this study (IRB No. 2410-023-1575), and the study adhered to the 1975 Declaration of Helsinki and its subsequent revisions.

We selected 226 patients with pancreatic cancer who underwent curative surgery after NAC at the Seoul National University Hospital between 2010 and 2020. Among the 226 patients, those with distant metastases (*n* = 37), a history of pancreas-related operations (*n* = 7), and those who underwent palliative operations (*n* = 23) were excluded. Additionally, only patients with ductal adenocarcinoma (PDAC) of the pancreas were included, excluding those with other pathologies (*n* = 11). This study focused on PDAC, the most common subtype of pancreatic cancer, as it accounts for the majority of cases. Other subtypes, such as intraductal papillary mucinous carcinoma, endocrine tumors, and acinar cell carcinoma, have different pathogeneses which could affect survival outcomes [40,41,42]. By excluding these subtypes, we wanted to specifically examine how nutritional changes impact survival in PDAC patients. Therefore, a total of 148 patients were included. The parameters of the CONUT score were assessed via a retrospective evaluation of laboratory values obtained at the time of initial diagnosis (before NAC) and right before surgery (about 3 weeks after NAC). Finally, patients were categorized into three groups based on changes in their CONUT scores before and after NAC: Worsened (*n* = 43, e.g., from normal to light or light to moderate), Maintained (*n* = 84, e.g., remained normal or light), Improved (*n* = 21, e.g., from light to normal or moderate to light) (Figure 1). This classification aids in understanding how NAC affects the immune–nutritional state of patients and highlights differences in nutritional status.

### 2.2. Data Collection and Follow-Up Protocol

Clinicopathological variables, including age, sex, NAC regimen, duration, and postoperative complications, were investigated. Additionally, information on the tumor location, operation type and method, lymph node metastasis, angiolymphatic invasion, venous invasion, perineural invasion, and histological grade was collected.

Patients were followed up in the outpatient clinic, where CA 19-9 levels and computed tomography (CT) scans of the chest, abdomen, and pelvis were routinely performed to monitor recurrence. Patient prognosis was evaluated by assessing the disease-free survival (DFS) and recurrence rates. OS was calculated as the interval between the date of surgery and the last follow-up or the date of death. DFS was defined as the duration between the date of surgery and the date of first recurrence, death, or last follow-up.

### 2.3. Statistical Analyses

Continuous variables were expressed as mean values with standard deviations, whereas categorical variables were presented as numbers with percentages. For continuous variables, Student’s *t*-test and analysis of variance were used, whereas categorical variables were compared using Pearson’s chi-square test and Fisher’s exact test. Pearson’s chi-square test was used to examine the differences between subgroups based on the CONUT score. The OS was measured from the time of surgery, and survival rates were calculated using the Kaplan–Meier method and compared using the log-rank test. In the analysis of OS sand DFS, we employed the Cox proportional hazards regression model, which is widely used in survival analysis to estimate the relative risk of an event, adjusting for potential confounders. This model allows for the identification of independent prognostic factors, ensuring that our findings are robust and accounting for potential biases in patient selection and treatment variation. A cox proportional hazard regression model hazard ratios (HR) with 95% confidence intervals (CI) were calculated. Variables with a *p*-value ≤ 0.1 in univariate analysis were included in multivariate analysis using Cox proportional hazards regression. Two-sided *p*-values < 0.05 were considered significant. The statistical power analysis was performed based on an assumed medium effect size (Cohen’s d = 0.5), a significance level of 0.05, and a desired statistical power of 80%. The calculated sample size required to detect meaningful differences was 64 participants per group. Given the actual patient population available, the final inclusion of 148 patients was sufficient to detect meaningful differences with adequate statistical power. IBM SPSS Statistics, Version 24 (IBM, Armonk, NY, USA) and the R software, version 4.4.2 (R Foundation) were used for all statistical analyses.

## 3. Results

### 3.1. Clinicopathologic Characteristics

The CONUT score is based on the concentrations of ALB, TLC, and TC. The mean ± SD values of the individual parameters before NAC were as follows: ALB: 4.0 ± 0.4 (g/dL), TLC: 1679.7 ± 801.6 (/mm^3^), and TC: 171.8 ± 43.9 (mg/dL). After NAC, the values were ALB: 3.8 ± 0.4 (g/dL), TLC: 1433 ± 698.9 (g/dL), and TC: 167.0 ± 43.3 (mg/dL). When comparing the values before and after NAC, both ALB and TLC showed a significant decrease (*p* < 0.001), while TC did not show a statistically significant change. The mean ± SD CONUT score before NAC was 1.9 ± 1.5, which increased to 2.5 ± 2.0 after NAC (*p* < 0.001) (Table 2).

According to the CONUT score, we assessed the nutritional status of patients and classified them into four groups (Table 1): normal, light, moderate, and severe. Before chemotherapy, the distribution was as follows: 42.6% in the normal group, 56.1% in the light group, and 1.4% in the moderate group, with no patients in the severe category. After NAC, the proportion of patients in the normal group decreased to 31.1%, while the number of patients in the light and moderate groups increased to 62.8% and 4.1%, respectively. Furthermore, the percentage of patients classified as severely malnourished also increased to 2%. This indicates a deterioration in nutritional status following NAC, as illustrated in Figure 2.

The Sankey diagram visualizes the transitions in nutritional status of pancreatic cancer patients before and after NAC. The diagram highlights the flow of patients between nutritional categories, with the width of each path corresponding to the number of patients shifting between groups. In our study, we observed a notable increase in the number of patients classified using CONUT score as moderately to severely malnourished after NAC, which aligns with previous studies suggesting that chemotherapy can adversely affect the nutritional status of patients.

In our study, based on the CONUT score, 148 patients were categorized into three groups based on their nutritional status change: 0–1 (normal), 2–4 (mild), and 5 or higher (moderate to severe): the ‘Worsened group’ (patients whose nutritional status worsened after NAC), the ‘Maintained group’ (patients whose nutritional status remained unchanged), and the ‘Improved group’ (patients whose nutritional status improved after NAC). The distribution of patients was as follows: Worsened group (*n* = 43, 29.1%), Maintained group (*n* = 84, 56.8%), and Improved group (*n* = 21, 14.2%).

The baseline characteristics of the study population are summarized in Table 3. Among the 148 patients, 74 (50.0%) were male, with a mean age of 61.5 ± 8.8 years. No significant difference in age was observed between male and female patients (*p* = 0.31).

Before NAC, the mean body weight and Body Mass Index (BMI) were 61.5 ± 9.2 kg and 23.2 ± 2.6 kg/m^2^, respectively. No significant differences in body weight were observed between the groups (*p* = 0.27), while BMI significantly differed (*p* = 0.03). The mean BMI was highest in the Improved group (24.6 ± 1.8 kg/m^2^), followed by the Maintained group (23.2 ± 2.8 kg/m^2^) and the Worsened group (22.6 ± 2.2 kg/m^2^). Regarding the CONUT score parameters, ALB levels were consistent across all groups (4.0 ± 0.4 g/dL, *p* = 0.94). However, TLC significantly differed among the groups (*p* < 0.001), with the highest mean TLC observed in the Worsened group (2075.1 ± 747.1 /mm^3^), followed by the Maintained group (1593.2 ± 789.2 /mm^3^) and the Improved group (1268.9 ± 661.2 /mm^3^). TC levels before NAC did not significantly differ between the groups (*p* = 0.27). The mean CONUT score before NAC was 1.9 ± 1.5, with significant differences observed among the groups (*p* < 0.001). The mean score was 2.8 ± 1.6 in the Improved group, 2.1 ± 1.4 in the Maintained group, and 1.1 ± 1.2 in the Worsened group. Similarly, PNI also differed significantly among the groups (*p* = 0.01). The mean PNI before NAC was 48.6 ± 5.8, with values of 50.8 ± 5.5 in the Worsened group, 48.1 ± 5.6 in the Maintained group, and 46.5 ± 6.2 in the Improved group. Since PNI is calculated using ALB and TLC, and ALB levels were identical across all groups, the observed differences in PNI were primarily influenced by variations in TLC among the groups.

Additionally, after NAC, ALB levels significantly differed among the groups (*p* < 0.001). The mean ALB was 3.6 ± 0.5 g/dL in the Worsened group, 3.9 ± 2.9 g/dL in the Maintained group, and 4.0 ± 0.3 g/dL in the Improved group. TLC also showed significant differences among the groups (*p* < 0.001). The mean TLC was 1056.6 ± 477.5 /mm^3^ in the Worsened group, 1512.8 ± 691.8 /mm^3^ in the Maintained group, and 1837.5 ± 813.2 /mm^3^ in the Improved group. Similarly, TC levels significantly differed among the groups after NAC (*p* < 0.001). The Worsened group had the lowest mean TC (149.4 ± 40.0 mg/dL), followed by the Maintained group (168.2 ± 41.1 mg/dL) and the Improved group (199.2 ± 44.7 mg/dL). Furthermore, the CONUT score increased significantly after NAC (*p* < 0.001). The Worsened group exhibited the highest CONUT score (4.1 ± 2.2), while the Maintained group had a mean score of 2.1 ± 1.5, and the Improved group had the lowest score (0.8 ± 1.4). PNI also significantly differed among the groups after NAC (*p* < 0.001). The mean PNI was 41.9 ± 5.8 in the Worsened group, 46.9 ± 4.5 in the Maintained group, and 49.4 ± 6.2 in the Improved group.

In terms of resectability, the resectability rate differed significantly between BRPC and LAPC cases (*p* = 0.04). The mean duration of NAC was also significantly longer in the Worsened group (5.0 ± 3.9 months) compared to the Maintained group (3.3 ± 3.1 months) and the Improved group (3.3 ± 1.9 months) (*p* < 0.001).

Other clinicopathologic factors, including tumor location, NAC regimen, operation type, and postoperative complications, did not show significant differences among the three groups (Table 3).

### 3.2. Survival Outcomes and Prognostic Factors for OS

We used the CONUT score to examine OS and DFS in order to evaluate our hypothesis that changes in nutritional status could affect patient prognosis. Before and after NAC, 148 patients were first categorized into three groups according to the severity of their malnutrition. The Kaplan–Meier survival curves for OS and DFS were then used to examine the survival results.

Before NAC, there were no significant differences in OS and DFS across the different levels of malnutrition as classified by the CONUT score. The median OS and DFS were similar between the normal, light, and moderate-to-severe malnutrition groups, with *p*-values of 0.93 and 0.49, respectively (Figure 3).

However, after NAC, we observed significant differences in OS and DFS among the malnutrition groups classified by the CONUT score. The moderate-to-severe malnutrition group had the shortest median OS of 11 months, followed by the light malnutrition group (36 months), while the normal group showed the longest median OS of 58 months (*p* < 0.001). A similar pattern was seen for DFS, with the moderate-to-severe group having the shortest median DFS of 8 months, followed by the light group (18 months), and the normal group with the longest median DFS of 35 months (*p* = 0.01, Figure 4).

These findings support previous research suggesting that the CONUT score, measured prior to surgery, is a significant prognostic factor for survival outcomes in patients with pancreatic cancer. Following these results, we conducted additional analyses to evaluate how changes in nutritional status, as reflected by the CONUT score’s change, influence both OS and DFS.

The 148 patients were divided into three groups according to changes in their nutritional status, in accordance with our original hypothesis: the “Worsened group” (those whose nutritional status deteriorated following NAC), the “Maintained group” (those whose nutritional status stayed constant), and the “Improved group” (those whose nutritional status improved after NAC). Significant differences in both OS and DFS were observed among these groups. Kaplan–Meier survival curves for each nutritional change group are presented in Figure 5. The overall median OS was 36 months, with the Worsened group exhibiting the shortest median OS (28, 39, and 66 months for the Worsened, Maintained, and Improved groups, respectively; *p* = 0.01). In addition, the overall median DFS was 18 months, and the Worsened group had the shortest median DFS (13, 22, and 39 months for the Worsened, Maintained, and Improved groups, respectively; *p* = 0.02, Figure 5).

Following the CONUT score analysis, we additionally examined two well-established factors to influence survival outcomes: BWL and the PNI.

BWL is a common and significant issue in patients with pancreatic cancer, and it often progresses over time. As previously discussed, up to 85% of patients with pancreatic cancer experience weight loss at the time of diagnosis, with approximately 50% showing preoperative weight loss and 80% experiencing progressive weight loss after diagnosis [14,15,16]. Notably, this progressive weight loss, which can be influenced by various factors, such as cancer-associated cachexia and malabsorption, has well-established predictive and prognostic implications for survival. The deterioration in nutritional status due to BWL also negatively impacts treatment tolerance, surgical outcomes, and overall survival [43,44,45]. As the disease progresses, ongoing BWL continues to present challenges in managing the nutritional needs of pancreatic cancer patients, highlighting the importance of timely interventions. Therefore, serial monitoring of body weight changes throughout treatment is crucial for understanding their impact on survival outcomes. In our study, we aimed to evaluate the impact of BWL on patient survival outcomes. The mean body weight of patients was 61.5 ± 9.2 kg before NAC and 61.3 ± 9.8 kg after NAC, with no statistically significant difference observed (*p* = 0.67, Table 4).

To further analyze the impact of body weight changes on survival, patients were categorized into three groups based on changes in body weight before and after NAC: the decreased group (≥5% weight loss), the maintained group (±5% weight change), and the increased group (≥5% weight gain). Among these groups, the decreased group significantly demonstrated the shortest median OS at 24 months, followed by the maintained group at 36 months. The increased group, which showed a significant weight gain of ≥5%, demonstrated the longest median OS of 66 months (*p* < 0.01, Figure 6).

As a well-established prognostic factor, the PNI has been widely used to assess immunonutritional status in patients with various cancers, including pancreatic cancer [26]. At first, it was suggested by Smale and colleagues in 1981, and, in this study, Onodera’s PNI was in used [46,47,48]. Originally proposed as a preoperative risk factor and determinant of surgical indication in colorectal cancer, this PNI is now widely used as a nutritional assessment barometer by hospitalized patient nutrition support teams as well as in the field of gastrointestinal surgery [48]. PNI plays a significant role in predicting postoperative complications and prognosis in patients with different types of tumors [34].

PNI is calculated using two key indices: serum albumin and total lymphocyte count, with the formula: PNI = 10 × ALB (g/dL) + 0.005 × TLC (/mm^3^).

Notably, these two parameters are also key components of the CONUT score, and low PNI values are associated with poorer survival outcomes and a higher rate of postoperative complications, particularly pancreatic fistula in patients with pancreatic cancer [26].

Given its prognostic value, we analyzed changes in PNI before and after NAC to evaluate its impact on survival outcomes in pancreatic cancer patients (Table 5). Previous studies have demonstrated that a PNI cutoff value of 45 effectively identifies patients with compromised nutritional and immune status, which correlates with worse postoperative outcomes [34]. In our study, the mean PNI after NAC was 45.9, which closely aligns with the previously established cutoff of 45 for predicting poor survival outcomes in patients with pancreatic cancer.

Comparing PNI values before and after NAC, we observed a significant decrease, reflecting a deterioration in nutritional and immune status following treatment.

We evaluated the impact of PNI on patient survival outcomes, categorizing patients into low and high PNI groups based on a cutoff value of 45. Before NAC, patients with low PNI showed a trend toward poorer survival outcomes; however, the differences were not statistically significant (Figure 7).

After NAC, the median OS was significantly shorter in the low PNI group (24 months) compared to the high PNI group (41 months, *p* = 0.02), suggesting that low PNI after NAC is associated with worse survival outcomes (Figure 8).

To further analyze the impact of PNI changes on survival outcomes, patients were categorized into four groups based on PNI before and after NAC: High→High, High→Low, Low→High, and Low→Low. Patients who maintained a high PNI (High→High) after NAC demonstrated the longest median OS (46 months), followed by those whose PNI increased (Low→High; 32 months). In contrast, patients whose PNI worsened (High→Low; 26 months) or remained low (Low→Low; 19 months) exhibited poorer survival outcomes, but the difference was not statistically significant (*p* = 0.13, Figure 9). Regarding DFS, patients who maintained a high PNI had the longest median DFS (35 months), while those who maintained a low PNI had the shortest median DFS (10 months) and the differences in DFS were statically significant (*p* = 0.04, Figure 9).

To further assess the factors influencing OS, univariate and multivariate regression analyses were conducted, with the results summarized in Table 6. In multivariate analyses, we identified that changes in nutritional status, particularly the Worsened group, were significantly associated with OS (HR 2.11; 95% CI, 1.31–3.40; *p* < 0.01). Additionally, R1 resection was a significant factor influencing survival outcomes (HR, 2.67; 95% CI, 1.44–4.94; *p* < 0.01). Other notable risk factors included diabetes (HR, 1.60; 95% CI, 1.03–2.48; *p* = 0.03), postoperative complications (HR 2.07; 95% CI 1.14–3.76; *p* = 0.01), and venous invasion (HR 1.65; 95% CI, 1.03–2.66; *p* = 0.03).

## 4. Discussion

We investigated the relationship between changes in nutritional status and OS in patients with pancreatic cancer who underwent curative surgery after NAC. As illustrated in Figure 2, the Sankey diagram highlights the transitions in nutritional status, showing a significant increase in the number of patients with moderate to severe malnutrition after NAC. These findings suggest that, similar to previous studies, chemotherapy tends to lead to nutritional deterioration. In our study, the chemotherapy regimens included FOLFIRINOX and Gemcitabine-based therapies, such as GemAbraxane (a combination of Gemcitabine and Nab-paclitaxel), with FOLFIRINOX accounting for 77% (114 patients) and Gemcitabine-based chemotherapy for 23% (34 patients) (Table 3). The duration of treatment was significantly longer in the Worsened group (5.0 ± 3.9 months) compared to the Maintained group (3.3 ± 3.1 months) and the Improved group (3.3 ± 1.9 months) (*p* < 0.001), suggesting that longer chemotherapy duration may affect the nutritional status of the patients. Especially FOLFIRINOX, a combination chemotherapy regimen consisting of oxaliplatin, irinotecan, fluorouracil, and leucovorin is highly potent but associated with significant hematological toxicity and gastrointestinal events. These side effects, including low appetite, diarrhea, mucositis, nausea, and vomiting, are associated with reduced food intake, impairing nutritional consumption, and potentially leading to weight loss and decreased protein intake.

As a tool for assessing nutritional status, the CONUT score is easily calculated using ALB, TLC, and TC. These parameters reflect protein stores (albumin), caloric depletion (cholesterol, as a marker of energy deficiency), and immunological defense (lymphocytes), respectively. The CONUT score provides a comprehensive assessment of immune–nutritional status, going beyond simple nutritional evaluation by simultaneously reflecting immune function and inflammatory status.

Serum albumin is widely recognized as an objective measure of nutritional status, but they can be influenced by acute inflammation, liver function, and other non-nutritional factors [49]. Protein-calorie malnutrition is attributed to poor nutritional intake and inflammation suppress albumin synthesis [50]. As part of the systemic inflammatory response to tumors, proinflammatory cytokines and growth factors are released by the tumor and surrounding cells, exerting significant catabolic effects on host metabolism [51,52,53]. Interleukin-6(IL-6), produced by the tumor or surrounding cells, stimulates the liver to produce acute-phase response proteins (such as CRP and fibrinogen), both in the fasted and fed states [53]. This increases the demand for certain amino acids, which, if insufficient in the diet, may be derived from the breakdown of skeletal muscle [54]. Moreover, hematological toxicities of FOLFIRINOX, such as febrile neutropenia and reduced white blood cell count, contribute to the reduction in serum albumin levels through proinflammatory cytokines like Tumor necrosis factor (TNF)-α, IL-6, and CRP, which modulate hepatocyte function [55].

Furthermore, pancreatic cancer itself can induce systemic inflammatory responses. TNF increases the permeability of the microvasculature, promoting transcapillary passage of albumin [56]. The presence of micrometastatic tumor cells in the liver may also activate Kupffer cells to produce a variety of cytokines (IL-1b, IL-6, TNF), which regulate albumin synthesis by hepatocytes [55]. Therefore, the chronic systemic inflammatory response in advanced cancer patients may progressively lead to nutritional and functional deterioration [57]. As demonstrated in our study, serum albumin, which carries twice the weight of the other two components in the CONUT score, like weight loss, can negatively affect patient survival outcomes [58].

Both FOLFIRINOX and Gemcitabine, like other chemotherapeutic agent, can directly affect immune cells, leading to cytotoxic effects, changes in cell differentiation and function, and disruptions in immune cell communication and signaling pathways. These chemotherapy-induced changes contribute to lymphocyte depletion, compromising immune surveillance and weakening the anti-tumor immune response. Such immune suppression can not only reduce the efficacy of cancer treatments but also exacerbate patient outcomes [59]. Additionally, chemotherapy agents can indirectly affect nutrient absorption and metabolism by altering gastrointestinal function, potentially exacerbating nutritional depletion. For example, nutritional markers such as triceps skinfold thickness and mid-upper arm circumference have been shown to correlate with TLC, reflecting the interplay between nutritional status and immune function [60]. In addition, a low peripheral lymphocyte count has been associated with poor prognosis in various types of cancer and serves as an indicator of inadequate host immune response [61].

Chemotherapy regimens not only induce gastrointestinal side effects and hematological toxicity, but can also affect lipid metabolism, leading to changes in cholesterol levels, which may reflect a deteriorating nutritional state. Cholesterol is influenced by inflammatory responses associated with chemotherapy, and a reduction in serum cholesterol levels may reflect a deteriorating nutritional state [62].

The higher predictive accuracy of the CONUT score compared to other immune–nutritional biomarkers may be attributed to its ability to comprehensively assess immune–nutritional status by combining these three variables. For each component that decreases, a higher score is assigned, indicating poorer nutritional status.

For these reasons, we assessed patients’ nutritional status before and after NAC using the CONUT score in our study. Before NAC, no statistically significant differences in OS and DFS were observed among the malnutrition groups (*p* = 0.93 and *p* = 0.49, respectively). However, after NAC, significant differences in OS and DFS were observed among the malnutrition groups classified by the CONUT score (*p* < 0.001 and *p* = 0.01). These findings suggest that initial nutritional status alone was not a prognostic factor for survival, but the impact of malnutrition on survival becomes more pronounced during NAC, possibly due to the cumulative effects of treatment-related nutritional decline. For this reason, when comparing patients longitudinally according to changes in their nutritional status, significant differences in both OS and DFS were observed (*p* = 0.01, *p* = 0.02). These findings indicate that nutritional deterioration after NAC is an independent prognostic factor for OS and DFS in patients with pancreatic cancer.

The concept of nutritional status is more complex than what the CONUT score alone can assess. To further explore the complexity of nutritional status, we conducted survival analyses based on body weight changes. Additionally, in our study population, we evaluated prognosis using PNI, a marker of nutritional status that incorporates ALB and TLC. This analysis was performed following the same approach as the CONUT score evaluation of nutritional status. Before and after NAC, the overall body weight of our study population did not show significant changes. However, when categorizing patients based on body weight changes (≥5 kg) before and after NAC, both OS and DFS demonstrated significant differences (*p* < 0.001, *p* < 0.01), similar to the results observed with the CONUT score. Furthermore, PNI, a well-established prognostic factor, was widely used in this study to assess changes in nutritional status before and after NAC. When stratifying patients into High PNI and Low PNI groups based on a cutoff value of 45, no significant differences were found in OS and DFS before NAC (*p* = 0.28 and 0.61), but, after NAC, significant differences were observed (*p* = 0.02 and *p* < 0.01). Moreover, when observing changes in nutritional status, the group that maintained good nutritional status (High→High) had the longest median OS, while the group whose nutritional status deteriorated (Low→Low) had the shortest median OS (46 months vs. 19 months). However, it was not statistically significant.

These findings suggest that the CONUT score alone, which is based on three parameters, has limitations in comprehensively assessing overall nutritional status. By incorporating body weight changes and PNI, this study provides a more comprehensive understanding of the impact of nutritional status changes on survival outcomes in patients with pancreatic cancer. While these additional markers help in evaluating nutritional status more holistically, they do not fully capture the complexity of nutritional health, highlighting the need for a broader range of biomarkers to assess nutritional status in clinical practice.

Numerous studies have highlighted the critical importance of maintaining adequate nutritional status in patients with cancer, as it significantly impacts their ability to tolerate treatments and influences tumor progression [25]. Consistent with our findings, mounting data have demonstrated that the nutritional status of cancer patients is a crucial determinant, as it can predict their ability to withstand therapy and tumor growth [63]. The detrimental effects of malnutrition on the quality of life and treatment hazards have been estimated to result in the death of up to 10–20% of cancer patients, primarily because of the repercussions of malnutrition rather than the tumor itself [64]. One of the determinant factors of malnutrition in patients with pancreatic cancer appears to be pancreatic exocrine insufficiency (PEI), which occurs due to obstruction of the main pancreatic duct, fibrosis of the gland, and loss of pancreatic exocrine tissue [65].The standard treatment for PEI is pancreatic enzyme replacement therapy (PERT) [66]. PERT has been well documented to lead to significant improvements in body weight and other nutritional parameters in pancreatic cancer patients, which, in turn, positively impacts survival outcomes [67,68,69,70]. This finding aligns with our study results, which underscore the critical role of nutritional status in influencing survival rates. Specifically, based on a retrospective cohort study carried out at our institution in 2019, over 87% of patients who underwent pancreaticoduodenectomy or pylorus-preserving pancreaticoduodenectomy had prior nutritional deficits associated with both postoperative morbidity and impaired quality of life [17]. Additionally, not only the pancreatic cancer tumor itself, but NAC also demonstrated a significant toxicity rate and may have a direct effect on nutritional status [71,72].

Pancreatic cancer patients often experience nutritional deficiencies due to the combined effects of the tumor, surgery, and NAC, all of which can impact survival rates. However, despite the importance of addressing nutritional status, there is currently a lack of comprehensive treatment approaches that focus on continuous nutritional management throughout the treatment process. Most previous studies have primarily focused on preoperative nutritional status in isolation.

A distinguishing feature of our study is the comprehensive evaluation of the sequential changes in nutritional status both before and after NAC, unlike most previous studies that primarily focused solely on the preoperative nutritional status only [73,74,75,76]. As a new immunonutrition marker, CONUT score has already been demonstrated in previous studies as an effective independent predictor of OS in patients with pancreatic cancer undergoing radical surgery [39]. In this study, we confirmed that changes in nutritional status, as assessed with the CONUT score, were independent prognostic factors for both OS and DFS. Our results revealed that nutritional deterioration during NAC is a significant prognostic factor, comparable to the surgical resection margin, in predicting survival outcomes. This finding indicates that it is crucial not only to evaluate nutritional status at a single time point but also to monitor changes over time. CONUT score is a simple blood test that allows easy measurement of a patient’s nutritional status, and it enables continuous monitoring of changes in nutritional status over time. Therefore, continuous monitoring of nutritional changes during NAC is essential, as it can significantly impact the therapeutic approach.

This study has some limitations. First, because this was a retrospective study, it was not possible to exclude the potential influence of uncontrolled confounding variables, such as chemotherapy regimen or individual treatment responses. Patients were divided into three groups based on changes in their nutritional status. The number of patients in each group varied, and smaller group sizes may have affected the reliability of the in-depth analysis. Further research using a controlled design is required to verify these results. Secondly, the specific causes of nutritional status changes during NAC, such as digestive disturbances, side effects of chemotherapy, or other complications, were not clearly identified. Unfortunately, our database was insufficient to determine the specific causes of nutritional changes. As mentioned earlier, the concept of nutritional status is complicated and multidimensional. Although the CONUT score offers important information about a patient’s nutritional and immunological health, it does not account for all variables that may affect nutritional status. Given this, recent advances in the study of the microbiome have brought attention to the possible involvement of the gut microbiota in the development and progression of pancreatic cancer, with growing evidence indicating a link between microbiota composition and cancer susceptibility [77]. Chemotherapy-induced changes in the gut microbiome may worsen nutritional malabsorption, increase inflammation, and further impair the absorption of nutrients [78]. The complex connection between nutrition and the microbiome highlights the need for more thorough biomarkers and models to evaluate patient health. In our ongoing study, we are investigating the relationship between pancreatic cancer and the microbiome, focusing on identifying potential early diagnostic markers. By analyzing fecal microbiota and metabolites in both patients with pancreatic cancer and their families, we aim to better understand the role of microbiome dysbiosis in the progression of pancreatic cancer. Preliminary findings suggest that certain microbial species may be associated with both the initiation and progression of pancreatic cancer, which could complement traditional nutritional markers in predicting treatment response and patient outcomes. The findings from this study, which incorporate metagenomics and metabolomics approaches, are expected to provide insights into personalized nutrition and therapeutic interventions tailored to the microbiome, offering a more holistic view of patient health during cancer treatment. Third, the analysis did not include additional variables of “prehabilitation” such as physical function and mental condition. The aim of “prehabilitation” is to alter metabolic risk by employing a threefold strategy that includes physiological activity, nutrition, and the reduction in psychological stress [79]. The implementation of prehabilitation enhanced the patient’s functional capacity and decreased postoperative complications [80,81]. Moreover, several studies have demonstrated favorable long-term results in specific patients [82]. Based on prior studies, it is crucial to use all three elements in prehabilitation [83]. However, this study did not assess the effects of exercise or mental stress.

However, despite these limitations, immediate attention to the integration of nutritional interventions into clinical treatment is crucial, and this study provides a foundation for future research and practical implementation of continuous nutritional management throughout treatment. While it is not surprising that nutritional deterioration after NAC is a significant prognostic factor of poor survival outcomes in pancreatic cancer patients, this finding underscores the critical need for effective nutritional interventions. Although the association between nutritional status and survival has been established in prior studies, our research offers further evidence in the context of NAC, highlighting the importance of longitudinal nutritional assessment. It is essential to evaluate the nutritional status of patients at the time of pancreatic cancer diagnosis and prior to the initiation of chemotherapy rather than concentrating only on preoperative nutritional assessment. Moreover, it is crucial to implement multimodal “prehabilitation” interventions to address other factors, such as exercise and anxiety reduction strategies, which are known to influence nutritional status [84]. This includes providing interventions and education aimed at improving nutritional status. Further studies are necessary to determine the impact of these interventions on improving nutritional status and, ultimately, on assessing their potential to enhance OS in patients with pancreatic cancer.

## 5. Conclusions

In conclusion, worsened nutritional status during NAC significantly impacts prognosis, similar to the effects of R1 resection. Despite its importance, nutrition has received insufficient attention. Consequently, the approach to managing nutritional status in pancreatic cancer patients must shift. Active management of nutritional status throughout treatment, including regular nutritional assessments, timely interventions such as PERT, multimodal prehabilitation programs, and the use of immunonutrition markers like the CONUT score, is crucial for improving patient outcomes. Our study suggests that maintaining, if not increasing, nutritional status during NAC may contribute to improved oncologic outcomes.

## Figures and Tables

**Figure 1 nutrients-17-00647-f001:**
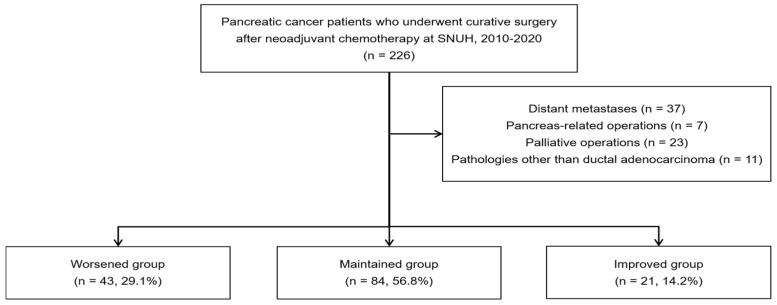
Flow chart of patient selection.

**Figure 2 nutrients-17-00647-f002:**
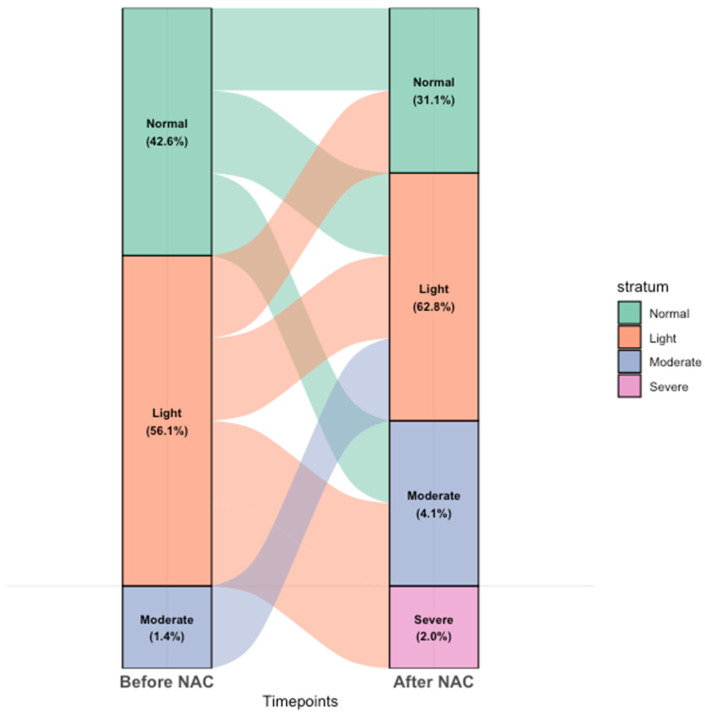
Sankey’s diagram of changes in nutritional status before and after NAC.

**Figure 3 nutrients-17-00647-f003:**
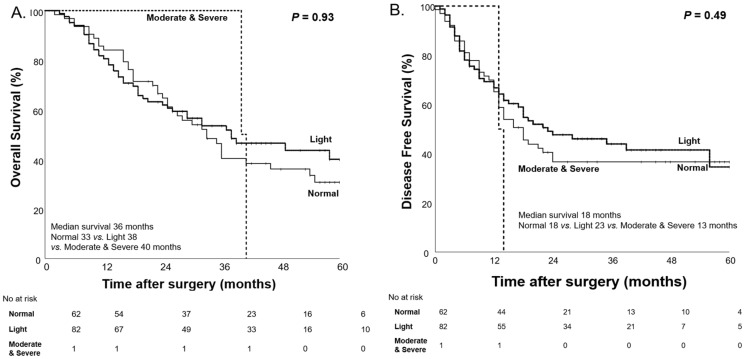
Kaplan–Meier curves for overall survival (**A**) and disease-free survival (**B**) based on the CONUT score before NAC.

**Figure 4 nutrients-17-00647-f004:**
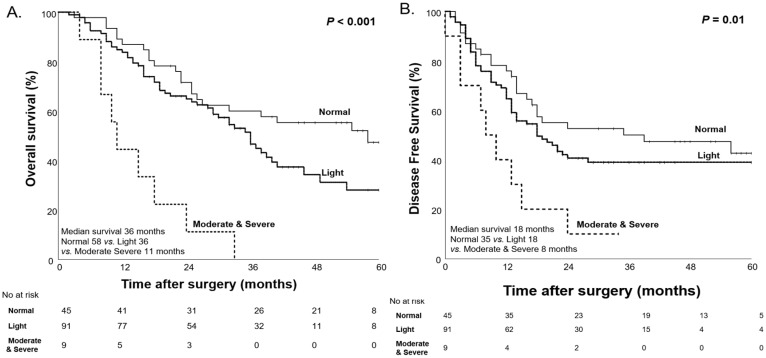
Kaplan–Meier curves for overall survival (**A**) and disease-free survival (**B**) based on the CONUT score after NAC.

**Figure 5 nutrients-17-00647-f005:**
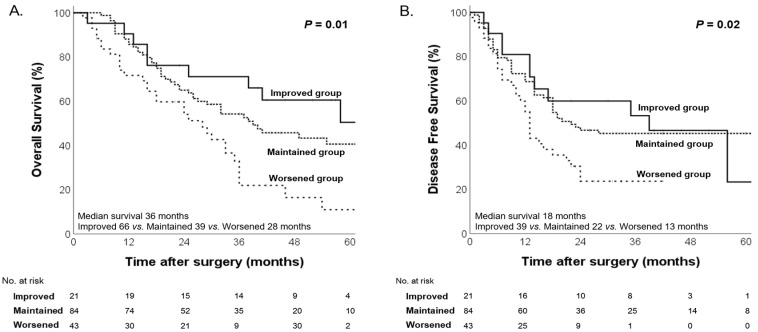
Kaplan–Meier curves for overall survival (**A**) and disease-free survival (**B**) according to changes in nutritional status.

**Figure 6 nutrients-17-00647-f006:**
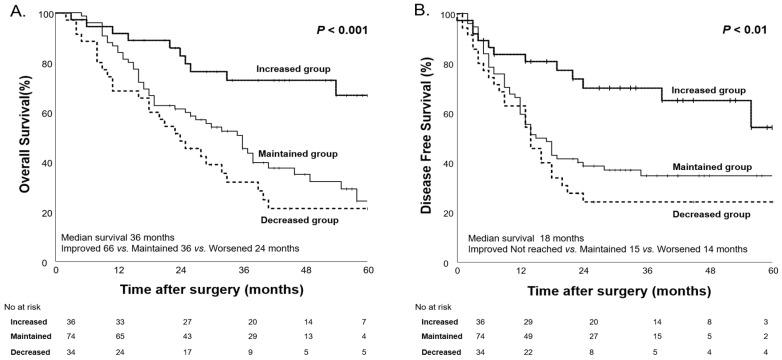
Kaplan–Meier curves for overall survival (**A**) and disease-free survival (**B**) based on body weight changes after NAC.

**Figure 7 nutrients-17-00647-f007:**
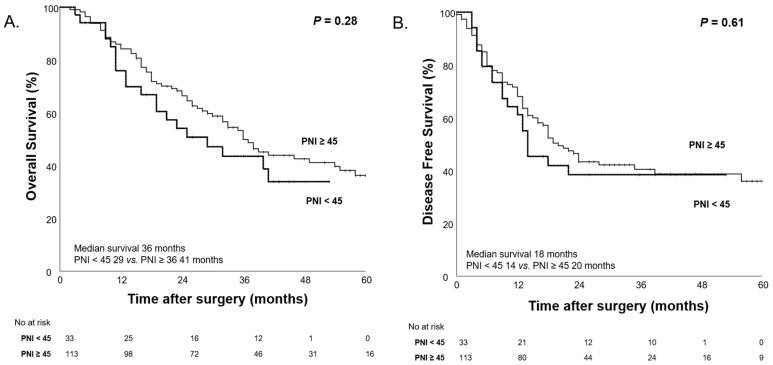
Kaplan–Meier curves for overall survival (**A**) and disease-free survival (**B**) according to high PNI (PNI ≥ 45) and low PNI (PNI < 45) before NAC.

**Figure 8 nutrients-17-00647-f008:**
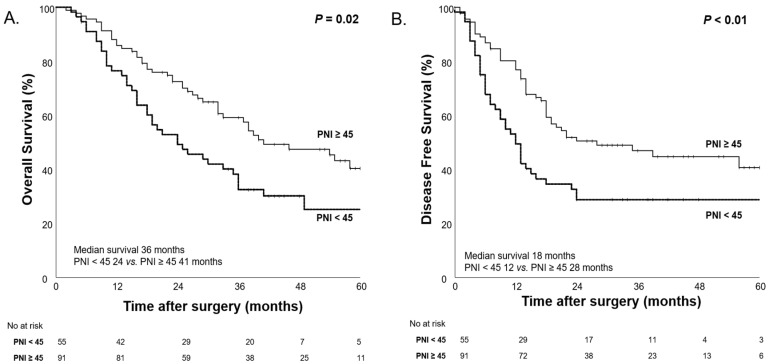
Kaplan–Meier curves for overall survival (**A**) and disease-free survival (**B**) according to high PNI (PNI ≥ 45) and low PNI (PNI < 45) after NAC.

**Figure 9 nutrients-17-00647-f009:**
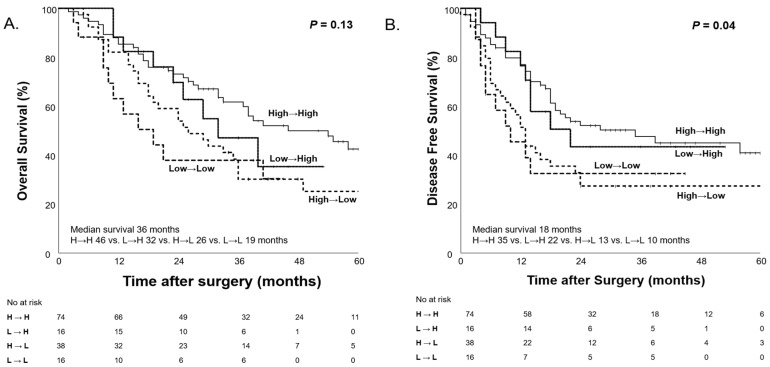
Kaplan–Meier curves for overall survival (**A**) and disease-free survival (**B**) according to PNI changes (Low PNI < 45, High PNI ≥ 45) before and after NAC.

**Table 1 nutrients-17-00647-t001:** Assessment of the nutritional status according to the controlling nutritional status (CONUT) score.

Parameters	Malnutrition Degree			
	Normal	Light	Moderate	Severe
Serum albumin (g/dL)	≥3.50	3.00–3.49	2.50–2.99	<2.50
Serum albumin score	0	2	4	8
Total lymphocyte count (/mm^3^)	≥1600	1200–1599	800–1199	<800
Total lymphocyte count score	0	1	2	3
Total cholesterol (mg/dL)	≥180	140–180	100–139	<100
Total cholesterol score	0	1	2	3
CONUT score = Serum albumin score + Total lymphocyte count score + Total cholesterol score				
CONUT score	0–1	2–4	5–8	9–12
Assessment	Normal	Light	Moderate	Severe

**Table 2 nutrients-17-00647-t002:** CONUT score parameters before and after NAC.

Variables	Before NAC	After NAC	*p*-Value
CONUT * score parameters			
ALB *, mean (SD), g/dL	4.02 (0.4)	3.8 (0.4)	<0.001
TLC *, mean (SD), /mm^3^	1679.74 (801.6)	1433.0 (698.9)	0.20
TC *, mean (SD), mg/dL	171.8 (43.9)	167.0 (43.3)	<0.001
CONUT score, mean (SD)	1.96 (1.5)	2.5 (2.0)	<0.001

* CONUT, controlling nutritional status; ALB, Serum Albumin; TLC, Total Lymphocyte Count; TC, Total Cholesterol; NAC, Neoadjuvant Chemotherapy.

**Table 3 nutrients-17-00647-t003:** Demographics of overall patients.

	Total	Worsened Group	Maintained Group	Improved Group	
Variables	(*n* = 148)	(*n* = 43)	(*n* = 84)	(*n* = 21)	*p*-Value
Age, mean (SD *), years	61.5 (8.8)	62 (6.9)	59 (9.7)	62 (8.0)	0.31
Sex (Male) (%)	74 (50.0)	19 (44.2)	45 (53.6)	10 (47.6)	0.58
Hypertension (%)	58 (39.2)	20 (46.5)	30 (35.7)	8 (38.1)	0.49
DM (%)	77 (52.0)	24 (55.8)	45 (53.6)	8 (38.1)	0.37
ASA * (%)					0.72
I	19 (12.8)	6 (14.0)	12 (14.3)	1 (4.8)	
II	106 (71.6)	32 (74.4)	57 (67.9)	17 (81.0)	
III	23 (15.5)	5 (11.6)	15 (17.9)	3 (14.3)	
Before NAC *					
Body weight, mean (SD),kg	61.5 (9.2)	59.8 (8.9)	61.7 (9.4)	64.1 (8.3)	0.27
BMI *, mean (SD), kg/m^2^	23.2 (2.6)	22.6 (2.2)	23.2 (2.8)	24.6 (1.8)	0.03
ALB *, mean (SD), g/dL	4.0 (0.4)	4.0 (0.4)	4.0 (0.4)	4.0 (0.4)	0.94
TLC *, mean (SD), /mm^3^	1679.7 (801.6)	2075.1 (747.1)	1593.2 (789.2)	1268.9 (661.2)	<0.001
TC *, mean (SD), mg/dL	171.8 (43.9)	181.2 (46.3)	169.5 (43.7)	163.7 (38.9)	0.27
CONUT * score, mean (SD)	1.9 (1.5)	1.1 (1.2)	2.1 (1.4)	2.8 (1.6)	<0.001
PNI, mean (SD)	48.6 (5.8)	50.8 (5.5)	48.1 (5.6)	46.5 (6.2)	0.01
After NAC					
Body Weight, mean (SD), kg	61.3 (9.9)	59.0 (9.6)	61.6 (9.9)	65.0 (9.0)	0.09
BMI, mean (SD), kg/m^2^	23.2 (3.0)	22.4 (2.7)	23.2 (2.8)	25.1 (3.5)	<0.01
ALB, mean (SD), g/dL	3.8 (0.4)	3.6 (0.5)	3.9 (2.9)	4.0 (0.3)	<0.001
TLC, mean (SD), /mm^3^	1433.0 (698.9)	1056.6 (477.5)	1512.8 (691.8)	1837.5 (813.2)	<0.001
TC, mean (SD), mg/dL	167.0 (43.3)	149.4 (40.0)	168.2 (41.1)	199.2 (44.7)	<0.001
CONUT score, mean (SD)	2.5 (2.0)	4.1 (2.2)	2.1 (1.5)	0.8 (1.4)	<0.001
PNI, mean (SD)	45.9 (5.6)	41.9 (5.8)	46.9 (4.5)	49.4 (6.2)	<0.001
Tumor location (%)					0.84
Head	105 (71.9)	28 (66.7)	61 (73.5)	16 (76.2)	
Body/Tail	37 (25.3)	12 (28.6)	20 (24.1)	5 (23.8)	
Diffuse	4 (2.7)	2 (4.8)	2 (2.4)	0 (0.0)	
Resectability (%)					0.04
BRPC *	118 (79.7)	31 (72.1)	73 (86.9)	14 (66.7)	
LAPC *	30 (20.3)	12 (27.9)	11 (13.1)	7 (33.3)	
NAC Regimen (%)					0.62
FOLFIRINOX *	114 (77.0)	32 (74.4)	64 (76.2)	18 (85.7)	
Gem + based	34 (23.0)	11 (25.6)	20 (23.8)	3 (14.3)	
Duration of NAC, mean (SD), month	3.8 (3.34)	5.0 (3.9)	3.3 (3.1)	3.3 (1.9)	<0.001
Operation type (%)					0.51
PD/PPPD *	100 (67.6)	26 (60.5)	58 (69.0)	16 (76.2)	
DP/SPDP*/STP *	36 (24.3)	13 (30.2)	18 (21.4)	5 (23.8)	
TP *	12 (8.1)	4 (9.3)	8 (9.5)	0 (0.0)	
Operation method (%)					0.18
OPEN	146 (98.6)	41 (95.3)	84 (100.0)	21 (100.0)	
MIS *	2 (1.4)	2 (4.7)	0 (0)	0 (0)	
Complication (%) (C-D * ≧ 3 grade)	21 (14.2)	7 (16.3)	10 (11.9)	4 (19.0)	0.60
Lymphatic invasion (%)	31 (20.9)	10 (23.3)	16 (19.0)	5 (23.8)	0.81
Venous invasion (%)	40 (27.0)	12 (27.9)	26 (31.0)	2 (9.5)	0.14
Perineural invasion (%)	106 (71.6)	30 (69.8)	63 (75.0)	13 (61.9)	0.46
R1 resection rate (%)	16 (10.8)	4 (9.3)	8 (9.5)	4 (19.0)	0.45
yp T stage (%)					0.92
T0	4 (2.7)	1 (2.4)	3 (3.6)	0 (0.0)	
T1	40 (27.2)	12 (28.6)	21 (25.0)	7 (33.3)	
T2	39 (26.5)	11 (26.2)	24 (28.6)	4 (19.0)	
T3	58 (39.5)	15 (35.7)	33 (39.3)	10 (47.6)	
T4	6 (4.1)	3 (7.1)	3 (3.6)	0 (0.0)	
yp N positive (%)	49 (33.1)	16 (37.3)	27 (32.2)	6 (28.6)	0.92

* SD, Standard Deviation; BMI, Body Mass Index; ASA, The American Society of Anesthesiologists physical status; ALB, Serum Albumin; TLC, Total Lymphocyte Count; TC, Total Cholesterol; CONUT, controlling nutritional status; NAC, Neoadjuvant Chemotherapy; BRPC, Borderline Resectable Pancreatic Cancer; LAPC, Locally Advanced Pancreatic cancer; FOLFIRINOX, Folinic acid (leucovorin), Fluorouracil, Irinotecan, and Oxaliplatin; Gem, gemcitabine; PD/PPPD, Pylorus Preserving/Pancreaticoduodenectomy; STP/SPDP/DP, Subtotal Pancreatectomy/Spleen-Preserving Distal Pancreatectomy/Distal Pancreatectomy; TP, Total Pancreatectomy; MIS, Minimally Invasive Surgery; C-D, Clavien–Dindo classification.

**Table 4 nutrients-17-00647-t004:** Changes in body weight before and after NAC.

Variables	Before NAC	After NAC	*p*-Value
Body weight, mean (SD), kg	61.5 (9.2)	61.3 (9.8)	0.67

**Table 5 nutrients-17-00647-t005:** PNI before and after NAC.

Variables	Before NAC	After NAC	*p*-Value
PNI *, mean (SD)	48.7 (5.8)	45.9 (5.6)	<0.001

* PNI, Prognostic Nutritional Index.

**Table 6 nutrients-17-00647-t006:** Univariate and multivariate regression analysis of prognostic factors for overall survival.

		Univariate Analysis			Multivariate Analysis		
Variables	Median OS [95% CI]	HR	[95% CI]	*p*-Value	HR	[95% CI]	*p*-Value
Sex (Male)	32 [25.80–38.19]	0.84	[0.55–1.28]	0.43			
Age (≥65)	29 [13.09–44.90]	1.48	[0.96–2.29]	0.07	1.41	[0.87–2.28]	0.16
DM (Yes)	26 [14.75–37.24]	1.58	[1.03–2.43]	0.03	1.60	[1.03–2.48]	0.03
Initial CA 19-9 >150 μ/mL	36 [24.69–47.30]	0.78	[0.51–1.20]	0.26			
Nutritional change	36 [29.29–42.70]			0.01			
Maintained Group	39 [24.17–53.82]			0.23			<0.01
Improved Group	66 [36.45–95.54]	0.83	[0.44–1.59]	0.59	0.87	[0.45–1.69]	0.70
Worsened Group	28 [17.00–38.99]	1.81	[1.13–2.89]	0.01	2.11	[1.31–3.40]	<0.01
Complication (C-D * ≧ 3 grade)	18 [12.47–23.52]	1.69	[0.95–3.01]	0.07	2.07	[1.14–3.76]	0.01
Adjuvant CTx * (Yes)	36 [29.44–42.55]	0.85	[0.26–2.71]	0.79			
RTx * (Yes)	38 [28.25–47.74]	0.60	[0.37–0.97]	0.03	0.76	[0.49–1.20]	0.24
Lymphatic invasion (Yes)	28 [8.32–47.67]	1.45	[0.89–2.35]	0.13			
Venous invasion (Yes)	23 [9.51–36.48]	1.68	[1.06–2.66]	0.02	1.65	[1.03–2.66]	0.03
Perineural invasion (Yes)	30 [21.78–38.21]	1.66	[1.00–2.76]	0.04	1.40	[0.80–2.44]	0.19
R1 resection (Yes)	19 [9.20–28.80]	1.97	[1.09–3.56]	0.02	2.67	[1.44–4.94]	<0.01
yp T stage	36 [29.16–42.83]			0.18			
T0	55 [1.21–108.78]			0.21			0.46
T1	40 [30.41–49.58]	1.77	[0.41–7.62]	0.44	1.01	[0.21–4.73]	0.98
T2	39 [22.47–55.52]	1.76	[0.41–7.57]	0.44	0.76	[0.15–3.56]	0.70
T3	26 [15.08–36.91]	2.42	[0.58–10.05]	0.22	1.18	[0.25–5.44]	0.83
T4	5 [0.00–12.20]	4.84	[0.88–26.53]	0.06	1.77	[0.23–13.29]	0.57
yp N stage	36 [29.29–42.70]			0.00			
N0	40 [26.02–53.97]			0.03			0.86
N1	32 [19.83–44.16]	1.38	[0.87–2.18]	0.16	1.07	[0.65–1.77]	0.77
N2	14 [7.55–20.44]	3.82	[1.50–9.68]	0.00	1.34	[0.43–4.20]	0.60

* C-D, Clavien–Dindo classification; CTx, Chemotherapy; RTx, Radiotherapy.

## Data Availability

Data are contained within the article.

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
