# Peer review of "Impact of Nutritional Changes on the Prognosis in Pancreatic Cancer Patients Underwent Curative Surgery After Neoadjuvant Chemotherapy"

_nutrients, 2025, doi:10.3390/nu17040647_

Round 1

Reviewer 1 Report

Comments and Suggestions for Authors

The manuscript submitted by Park et al., describes the findings of a human study that aimed to investigate the relationship between changes in nutritional status and OS in patients with pancreatic cancer who underwent curative surgery after NAC.

The manuscript is well organized and logically developed.

The reviewer has the following points for the authors to consider:

1. What was the rationale for the sample size? Was there a statistical analysis (power calculation) in terms of determining the number of participants?

2. The nutritional status is a more complex phenomenon than what the CONUT score entails. It would be important to discuss that aspect.

3. Have the authors considered drug nutrient interactions in their assessment and interpretation of the CONUT score?

4. Did the authors consider the microbiome? Perhaps a short discussion paragraph would be something to consider adding.

Reviewer 2 Report

Comments and Suggestions for Authors

In this manuscript/work, Park et al evaluated the effect of changes in nutritional status on the long-term outcomes of patients with pancreatic cancer who underwent curative surgery after neoadjuvant chemotherapy. For this, authors performed a retrospective single-center study with 148 patients with pancreatic cancer who underwent curative surgery after neoadjuvant chemotherapy between 2010 and 2020 using the Controlled Nutritional Status (CONUT) score to determine the nutritional status of the patients.

-the main results are not a surprise: nutritional deterioration after neoadjuvant chemotherapy is a significant prognostic factor of poor survival outcomes in patients with pancreatic cancer. Authors must comment on this in the manuscript

-“Although the nutritional impact of cancer treatment is well-known, few studies have specifically investigated how changes in nutritional status before and after…” – in the manuscript authors must reference here the “few studies” and demonstrate the novelty of the present work in relation to these few studies

-“CONUT score = Serum amylase score + Total lymphocyte count score + Total cholesterol score” – why amylase?

-“We selected 226 patients with pancreatic cancer who underwent curative surgery after NAC at the Seoul National University Hospital between 2010 and 2020. Among the 280 patients, those…” – 226 or 280? This must be clarified in the manuscript

-“only patients with ductal adenocarcinoma of the pancreas were included” – Why? This must be explained in the manuscript

Reviewer 3 Report

Comments and Suggestions for Authors

The manuscript by Park et al. comprehensively investigates the impact of changes in nutritional status (assessed with the CONUT score) before and after neoadjuvant chemotherapy on the outcome of patients with pancreatic cancer. The study design is retrospective. The main result is that a worsened nutritional status significantly predicts a poor outcome in these oncological patients. The background information provided in the introduction and the description of methods and findings are, in my opinion, satisfactory. Language, number of references, and quality of graphs are acceptable. Minor errors are present in the tables.

Since the manuscript is well-written, I will limit myself to minor comments.

Minor remarks

Page 2, line 74. I did not understand why 280 patients were mentioned if, in the flowchart in Figure 1, the initial patients were only 226. Please explain.

Page 3, Figure 1. In the top right block, preferably write “distant metastases” instead of “distant metastasis”.

Page 3, lines 97-107. The statistical section did not mention the use of Kaplan-Meier curves, although they were used in figure 2 (see line 126). If possible, also specify which statistical test was used in association with the Kaplan-Meier curves (log-rank?).

Page 3, lines 111-112. The percentages in brackets have been inverted (n = 21, 14.2%) and (n =43, 29.1%). Moreover, if possible, standardize the decimals of the “maintained group,” which is 56.8% in Figure 1 but 56.7% in the text (line 111).

Page 4, Table 2. The reported p-value for resectability between LAPC and BRPC cases is 0.03. However, based on the results of the Fisher exact test, it would be 0.038, so either this value is reported, or it should be approximated to 0.04.

Page 5, line 139. The upper limit of the confidence interval of hazard ratio is 3.79 in the text, but in Table 3 on page 6 it is 3.76. Please be consistent.

Round 2

Reviewer 1 Report

Comments and Suggestions for Authors

The authors have made reasonable efforts to address reviewer's points. Proofreading is suggested.

Reviewer 2 Report

Comments and Suggestions for Authors

The document was improved and now it is more acceptable for publication.